# Reconciling Geospatial Prediction and Retrieval via Sparse Representations

**Yi Li**
College of Computing and Data Science
Nanyang Technological University
liyi0067@e.ntu.edu.sg

**Yuanlong Chen**
College of Computing and Data Science
Nanyang Technological University
yuanlong001@e.ntu.edu.sg

**Weiming Huang**
School of Geography,
University of Leeds
W.Huang@leeds.ac.uk

**Xiaoli Li**
Institute for Infocomm Research, A*STAR
College of Computing and Data Science
Nanyang Technological University
xlli@i2r.a-star.edu.sg

**Gao Cong**[*]
College of Computing and Data Science
Nanyang Technological University
gaocong@ntu.edu.sg

## Abstract

Urban computing harnesses big data to decode complex urban dynamics and revolutionize location-based services. Traditional approaches have treated geospatial prediction tasks (e.g., estimating socio-economic indicators) and retrieval tasks (e.g., querying geographic objects) as isolated challenges, necessitating separate models with distinct training objectives. This fragmentation imposes significant computational burdens and limits cross-task synergy, despite advances in representation learning and multi-task foundation models.

We present UrbanSparse, a pioneering framework that unifies geospatial prediction and retrieval through a novel sparse-dense representation architecture. By synergistically combining these tasks, UrbanSparse eliminates redundant systems while amplifying their mutual strengths. Our approach introduces two innovations: (1) Bloom filter-based sparse encodings that compress high-sparsity geographic queries and fine-grained text terms for retrieval effectiveness, and (2) a dense semantic codebook that captures granular urban features to boost prediction accuracy. A two-view contrastive learning mechanism further bridges urban objects, regions, and contexts. Experiments on real-world datasets demonstrate 25.16% gains in prediction accuracy and 20.76% improvements in retrieval precision over state-of-the-art baselines, alongside 65.97% faster training. These advantages position UrbanSparse as a scalable solution for large urban datasets. To our knowledge, this is the first unified framework bridging geospatial prediction and retrieval, opening new frontiers in data-driven urban intelligence.[2]

---

[*]Corresponding author.

[2]Data and code available at https://github.com/pkuliyi2015/UrbanSparse

39th Conference on Neural Information Processing Systems (NeurIPS 2025).

# 1 Introduction

Over the past decade, we have witnessed a surge of urban data from a variety of sources, e.g., remote sensing images, points of interest (POIs), and human trajectories. This presents unprecedented opportunities for developing data-driven solutions to address various long-standing challenges, where various machine learning models have been developed for many tasks, such as economic growth prediction [26], air quality analysis [72], transport planning [7], and trajectory search [64]. Such tasks fall under two categories: *prediction* and *retrieval*.

Prediction tasks, also known as Geospatial Predictions [45], estimate holistic urban socio-economic indicators, either from data-rich areas to unknown areas or from the past to the future [38]. Representative tasks in this strand include predicting land use, population density, crime rates, and transportation [1, 24, 25, 33, 40, 45, 68]. Retrieval tasks, also known as Geographic Information Retrieval (GIR) [5, 32, 42, 58] focus on identifying relevant geographic entities by considering both keyword relevance and geographic proximity. For example, users searching for "coffee" along with GPS coordinates receive a list of nearby coffee shops. In this process, a GIR model computes the *relevance scores* between user queries and geographic objects to determine the order in which results are displayed to users. The effectiveness of retrieval can be evaluated and enhanced using *labeled queries*, where each query is labeled with one or more user-selected geographic objects. Prediction and retrieval tasks have traditionally been studied independently, driven by the long-held assumption that they require fundamentally different features (i.e., retrieval tasks emphasize low-frequency text terms [56], whereas prediction or classification tasks prioritize common or aggregated features [44]). With the development of representation learning and multi-task methods in both domains [1, 14, 27], which train one foundation model for multiple downstream tasks, it naturally leads us to a critical question: **can we develop a unified model to tackle and enhance both geospatial prediction and retrieval tasks?**

In this work, we reveal the great complementary advantages of jointly tackling the two tasks. For example, conventional prediction methods usually rely on POI density to estimate population density, which may fail in regions with a few large residential buildings (each suggesting a high number of residents). A unified model can utilize the abundant search queries from these residents to improve predictions. Likewise, traditional retrieval sorts geographic objects solely by fine-grained linguistic similarities and geographic proximity [18, 21, 39], whereas a unified model considers POI associations across the whole city and recommends similar areas that may meet a user's needs.

Despite these potential benefits, reconciling the inherent conflicts between the two tasks poses significant challenges. The first challenge is to *preserve fine-grained textual features*. Existing methods for prediction tasks generally use region-level data aggregation to extract geospatial proximity [50, 59] and regional associations [17, 70]. Such aggregation can ruin the textual details, leading to poor retrieval effectiveness. The second challenge is to *extract predictive information* from various text terms. While some distinctive, low-frequency text terms for retrieval tasks enhance predictions, many (e.g., "Postcode 101011") don't have semantics and may introduce noise or outliers into predictions. The third challenge is to *leverage labeled queries*. Though studies [14, 29, 39] demonstrate that fine-tuning on labeled queries improves retrieval performance, achieving such improvements in prediction tasks remains non-trivial and unexplored. Finally, existing prediction models require capturing complex spatial relationships, and retrieval methods often involve fine-tuning large language models, both face efficiency challenges on large datasets.

To address these challenges, we propose UrbanSparse, a unified framework for geospatial prediction and retrieval that employs a two-view learning mechanism capturing both fine-grained textual details and holistic regional context. First, in *Individual View*, we preserve fine-grained features by splitting texts with multiple tokenizers and encoding them as Bloom filter bits, evaluating and recording term-level importance with neural networks to mitigate information loss. Second, in *Collective View*, we maximize mutual information between regions and their geographic context to extract key predictive features while filtering out noise. Third, both views share a dense codebook trained with a novel warm-up strategy: we start with prediction tasks and then interleave training on both tasks, ensuring a smooth task transition. Finally, we introduce row- and column-selection techniques that leverage Bloom filter sparsity to boost efficiency. Experiments on real-world datasets demonstrate that UrbanSparse outperforms state-of-the-art baselines, achieving up to 25.16% improvement in prediction and 20.76% in retrieval effectiveness while reducing training time by 65.97% and memory requirements by 86.49% compared to traditional BERT-based embeddings.

In summary, our contributions are at least threefold:

- **A Novel Research Problem**: We tackle the critical yet underexplored challenge of unifying geospatial prediction and retrieval within a single framework. By showing that these traditionally separate tasks can be co-optimized for mutual benefit, our work paves the way for next-generation geospatial foundation models.

- **A Two-View Learning Mechanism**: We propose a two-view learning process that combines Bloom filter-based sparse representations for fine-grained textual encoding with graph contrastive learning for local and contextual geographic encoding. By maximizing the mutual information between regions and their surroundings, our approach learns useful features from Bloom filter bits without extensive pre-training.

- **Comprehensive Performance and Efficiency Gains**: Extensive experiments validate the superiority of UrbanSparse over state-of-the-art baselines, demonstrating significant improvements in effectiveness and efficiency. This framework not only advances task performance but also establishes a new standard for scalable and resource-efficient urban computing solutions.

## 2  Related Work

### 2.1  Geospatial Predictions

Geospatial predictions aim to estimate key urban characteristics by leveraging statistics and associations across urban regions. Early works like Yuan et al.[65] analyzed human mobility and POIs to identify functional zones, while Zheng et al.[72] combined meteorological data, road networks, and taxi movements to predict air quality. Street-level imagery has been used to assess urban safety [49], and social media data has uncovered urban patterns [16]. POI and check-in data are also used to classify urban zones [63]. However, these task-specific models lack generalizability.

Recent work explores unsupervised methods to get rid of task-specific labels and learn generalized urban representations. Wang et al.[60] introduced mobility graphs with human movements as edges, while Fu et al.[17] enhanced these with POIs via graph auto-encoders. Zhai et al.[67] modeled POI co-occurrence, and Niu and Silva[50] incorporated spatial proximity. Recent approaches include adversarial training for multi-modal data [70], attention mechanisms for cross-modal features [69], contrastive learning on multi-view [68] or hierarchical graphs [24] aggregation, and pre-trained foundation models [1, 27] for general-purpose region embeddings. However, these methods prioritize holistic features and neglect fine-grained details, unfeasible for retrieval tasks. In contrast, our method preserves both granular textual terms and holistic geospatial correlations to support both tasks.

### 2.2  Geographic Information Retrieval

Geographic Information Retrieval (GIR) handles text-based queries with geographic distances [52]. Early research [6, 10–12, 43] use a linear combination of distances [54] and unsupervised text similarities including TF-IDF [57] and BM25 [53] to identify relevant objects for queries. These methods use bag-of-words (BOW) to represent texts, which lacks deep semantics and limits their retrieval effectiveness and prediction accuracies. The integration of deep learning models into GIR marked a significant shift, using dense representations to compute text similarities. Early learning-based models [66, 71] encode texts with lightweight neural networks, yet they lack inherent semantic knowledge and rely on extensive labeled queries. Some text-based IR models [18, 21] are adapted to GIR tasks, which use Word2Vec [48] embeddings to estimate semantic relevance. However, empirical studies [39] demonstrate that these approaches [18, 21, 51] fail to compete with classical methods in retrieval effectiveness. Moreover, these methods don't generate reusable representations and struggle on large-scale GIR databases. Pre-trained Language Models (PLMs) like BERT [13] and ERNIE [22] advanced GIR by embedding semantic understanding. DrW [39] aligned BERT-based representations with query-aware geographic preferences, while MGeo [14] and ERNIE-GeoL [22] pre-trained on city-specific datasets, integrating mobility data and multi-modality features. Despite strong retrieval performance, these methods lacked holistic urban representations needed for prediction tasks and were computationally intensive. Our approach bridges this divide by learning individual-level relevance from labeled queries while incorporating collective-level urban context to support predictions.

## 3 Preliminaries

**Definition 1** (Representation Learning for Geographic Information Retrieval). *Given a spatial keyword query q and a geo-textual object o, each with a location and text, representation learning encodes their texts into vectors $f(q), f(o) \in \mathbb{R}^d$ so that*

$$Relevance(q, o) = F\big(DistSim(q, o), TextSim(f(q), f(o))\big),$$

*where DistSim and TextSim measure spatial proximity and text similarity, and F combines them. The top-k relevant objects are then retrieved from $S = \{o_1, \ldots, o_N\}$.*

**Definition 2** (Representation Learning for Geospatial Predictions). *Given an urban region $u \in \mathcal{U}$, representation learning encodes its spatial, functional, and containing objects into $f(u, o) \in \mathbb{R}^k$. A predictor $g : \mathbb{R}^k \to \mathbb{R}$ then estimates attributes $Y(u)$ (e.g., socio-economic indicators).*

**Problem Statement.** Given geo-textual objects $S = \{o_1, \ldots, o_N\}$ and regions $\mathcal{U} = \{u_1, \ldots\}$, learn a unified mapping $f$ such that

$$f(q), f(o) \in \mathbb{R}^d, \quad f(u) \in \mathbb{R}^k,$$

where $d$ and $k$ are the dimensions for queries/objects and regions, respectively.

## 4 Method

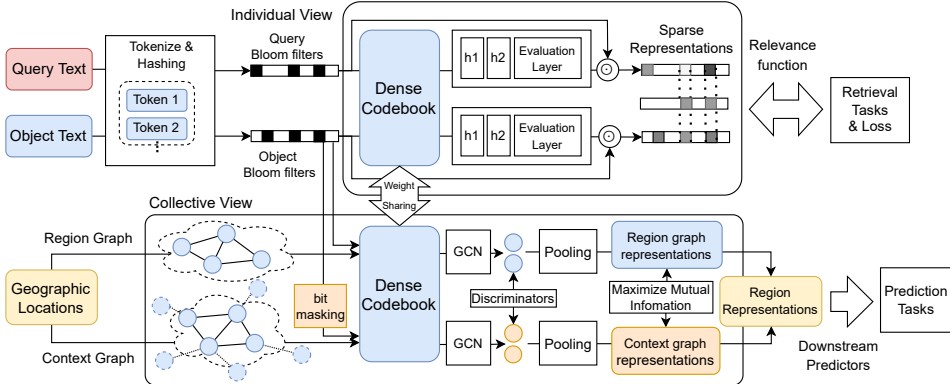

Figure 1: Overview of the UrbanSparse framework. (1) The *Individual View* encodes query and object text into Bloom filters, transforming them into sparse representations with fine-grained text importance. (2) The *Collective View* maximizes the mutual information between regions and their context, thereby learning meaningful geographic associations from Bloom filters.

We present UrbanSparse, a unified framework for geospatial prediction and retrieval that addresses four key challenges: (1) preserving fine-grained textual details for retrieval, (2) learning effective urban associations for prediction, (3) leveraging labeled queries for mutual task enhancement, and (4) improving computational efficiency. As shown in Figure 1, our system processes textual queries, geographic objects, and their spatial locations through two complementary views: First, the *Individual View* encodes text terms into Bloom filters via multiple hash functions. A neural network converts them into weighted sparse representations, avoiding information loss of dense vector aggregation. Second, the *Collective View* models urban regions through dual graph contrastive learning: an urban region graph capturing spatial proximity and a context graph encoding broader spatial influences. The mutual information maximization between regions and their contexts extracts geographic associations and filters noise. Third, a shared codebook connects the two view, with a warm-up training strategy for seamless task fusion. In addition, we propose row- and column-selection techniques that exploit Bloom filter sparsity to improve computational efficiency.

### 4.1 Text Encoding with Bloom filters

Textual descriptions of geographic objects are crucial for geospatial retrieval and prediction tasks. Traditional retrieval methods like BM25 use Bag-of-Words (BOW) representations, which effectively

identify key terms but struggle with scalability due to the vast vocabulary of geographic texts. Most prediction methods employ one-hot encoding of object categories [24, 33, 68], which neglect fine-grained text information. While recent pre-trained language models (PLMs) [1, 14, 29, 39] have improved semantic understanding, they incur high computational costs and slow inference speeds.

We observe that geographic texts contain distinctive city-specific terms like addresses, landmarks, and local business names. PLM-based approaches often underperform on these terms as they appear infrequently in pre-training corpora, requiring substantial fine-tuning data to match classical methods' performance (Table 4). This suggests deep semantic understanding from PLMs may be unnecessary, and that representing city-specific terms through finer lexical granularity could suffice. We therefore propose to decompose texts with multiple n-gram and dictionary-based tokenizers and then encode them to Bloom filters [2] via random hashing. This strategy preserves critical lexical patterns similar to BOW while maintaining constant dimensionality for efficient computation. We use vanilla Bloom filters in implementation, detailed in Appendix A.

Two key concerns arise in using Bloom filters: (1) *Hash Collision*, which means that distinct text terms may sometimes share the same hash value, leading to inaccurate representations. However, we empirically find that such collision doesn't significantly affect the holistic urban features important for predictions. In retrieval tasks, we can leverage a membership test (as in in Appendix A) to exclude query terms not present in the object's Bloom filters, reducing the negative impact of hash collisions. (2) *Loss of Token Order*, as Bloom filters only record the existence of terms without capturing their order. This may lead to inaccuracies in handling order-sensitive queries in retrieval tasks (e.g. treating "Unit 123456" and "Unit 654321" as the same term). However, the spatial keyword queries handled in GIR tasks are typically concise, and the n-gram tokenizer (as in DSSM [23]) is generally sufficient. A 3-gram tokenizer will turn "123456" into "12#", "#34#", "#56", encoding the local order of tokens.

## 4.2 Relevance Learning with Neural Networks

We then handle retrieval tasks with neural networks, computing a *relevance score* to determine the order in which results are displayed to users, as in Definition 1. We empirically found that methods based on term-matching can achieve strong effectiveness without training (e.g., BM25-D in Table. 4), as they effectively match city-specific terms without prior knowledge. Hence, we propose to leverage the inherent term-matching capabilities of Bloom filters, i.e., the non-zero bits in each Bloom filter correspond to text terms from the query or objects. Specifically, we *keep the representation sparse*, performing a bit-by-bit evaluation on Bloom filter bits:

$$\text{TextSim}(q, o) = (B_q \odot F_q(B_q)) \cdot (B_o \odot F_o(B_o)) \tag{1}$$

In Eq. 1, $B_q$ and $B_o$ are the Bloom filters (length $m$) for the query and object, respectively. We compute two sparse representations by reweighting the bits within the query and object Bloom filters separately. As in Figure 1, the encoders $F_q$ and $F_o$ contain a large, shared codebook matrix to encode the Bloom filters into dense embeddings, followed by two non-linear hidden layers that further compress them into low-dimensional space, extracting potential semantic relevance. Finally, the embeddings are expanded back via an *Evaluation Layer* to the dimension of the input Bloom filters, where each dimension is regarded as the importance of the bit at the corresponding position. The object Bloom filters can be evaluated offline, reducing online computations. The neural network $F(B)$ can be defined as:

$$\mathbf{h}_1 = cb(B) = \sigma(W_c B / \sum_i^m B_i), \quad W_c \in \mathbb{R}^{h_1 \times m} \tag{2}$$

$$\mathbf{h}_2 = \sigma(W_2 \mathbf{h}_1 + b_2), \quad W_2 \in \mathbb{R}^{h_2 \times h_1} \tag{3}$$

$$\mathbf{h}_3 = \sigma(W_3 \mathbf{h}_2 + b_3), \quad W_3 \in \mathbb{R}^{h_3 \times h_2} \tag{4}$$

$$F(B) = \sigma(W_4 \mathbf{h}_3) + 1, W_4 \in \mathbb{R}^{m \times h_3} \leftarrow 0 \tag{5}$$

Here, $\mathbf{h}_i$ denotes the output of i-th layer with dimension $h_i$, weight $W_i$, and bias $b_i$. $\sigma$ denotes the activation function. $W_c$ is the codebook matrix shared between the query and object encoder $F_q$ and $F_o$, and $\sum_i^m B_i$ is the number of non-zero bits within the Bloom filters, ensuring a consistent $h_1$ across the varying amount of text terms. Eq 5 is an evaluation layer tailored for Bloom filters, where $W_4 \in \mathbb{R}^{m \times h_3}$ is set to zero, ensuring each intersecting bit has equal initial importance of 1.

This preserves the term-matching capabilities of Bloom filters, which gives a good starting point in optimization that leads to faster convergence. Finally, we normalize and combine the text similarities with geographic distances:

$$T(q, o) = \text{Sigmoid}\left(\beta_1 \text{TextSim}(q, o) + \beta_2\right) \tag{6}$$

$$D(q, o) = -\log(1 + \text{Dist}(q, o)) \tag{7}$$

$$Relevance(q, n) = T(q, n) + \gamma_1 D(q, n) + \gamma_2 T(q, n) D(q, n) \tag{8}$$

Here, we use the logarithm function to align the distance with human spatial perceptions, i.e., individuals are more sensitive to differences in proximity with nearby objects, while this sensitivity diminishes for objects further apart. The normalization of the text similarities facilitates its smooth combination with geographic distances. $\beta_1, \beta_2, \gamma_1, \gamma_2$ rescale and balance the influence of two similarities and their first-order interaction, which better excludes proximate objects with little text similarities. We initially set $\beta_2 = \gamma_2 = 0$ and $\beta_1 = \gamma_1 = 1$, and train these parameters together with the neural networks via LambdaRank [3] loss.

### 4.3 Extracting Collective Features

Our objective is to learn geospatial associations critical for prediction tasks. Following most geospatial models [9], we employ Graph Neural Networks (GNNs) to preserve POI spatial relationships [8]. However, common self-supervised approaches like graph reconstruction struggle with Bloom filters, as they inherently mix informative text terms with useless terms. We posit that informative terms are those shared across regions but exhibit diverse spatial distributions. Unique terms like "Postcode 114514" are unhelpful because they only exist in one place and cannot be leveraged by downstream predictors. The density of useful terms like "Starbucks" helps identify commercial zones.

Inspired by Tobler's Second Law of Geography ("the phenomenon external to a geographic area of interest affects what goes on inside"), we learn useful information from Bloom filters by maximizing the mutual information between city regions and their surroundings. Specifically, we construct a city-wise graph with Delaunay Triangulation following [24, 33] and perform contrastive learning on two graphs following [20, 73]. The two graphs include: 1) A region graph consisting of objects within a region, which captures intra-region bit distribution patterns, and 2) A context graph incorporating K-hop neighborhoods of the region graph. We utilize two 2-layer Graph Convolutional Networks (GCNs) [30] to compute object-level and graph-level representations:

$$Z_o^r = \text{MLP}_1\left(\sigma(A_r H_r W_1^r + b_1^r)W_2^r + b_2^r\right), \quad Z_g^r = \text{MLP}_2\left(\text{AvgPool}(Z_o^r)\right), \tag{9}$$

$$Z_o^c = \text{MLP}_1\left(\sigma(A_c H_c W_1^c + b_1^c)W_2^c + b_2^c\right), \quad Z_g^c = \text{MLP}_2\left(\text{AvgPool}(Z_o^c)\right). \tag{10}$$

where $A_r, A_c$ are adjacency matrices, $H_r, H_c$ are the features from the dense codebook in Eq. 2, $W_i^r, W_i^c$ are learnable weights, and $b_i^r, b_i^c$ are biases in the i-th GCN layer. AvgPool denotes the graph-level average pooling. We then maximize the mutual information (MI) between the region and context graphs, defined as:

$$\mathcal{L}_{\text{pred}} = -\frac{1}{|\mathcal{V}|} \sum_{o=1}^{|\mathcal{V}|} \left\{ \text{MI}(Z_o^r, Z_g^c) + \text{MI}(Z_o^c, Z_g^r) \right\}, \tag{11}$$

where the MI estimation is computed as:

$$\text{MI}(Z_o^r, Z_g^c) = \mathbb{E}_{r,c}\left[\log D(Z_o^r, Z_g^c)\right] + \mathbb{E}_{\hat{r},c}\left[\log\left(1 - D(\hat{Z}_o^r, Z_g^c)\right)\right],$$

$$\text{MI}(Z_o^c, Z_g^r) = \mathbb{E}_{c,r}\left[\log D(Z_o^c, Z_g^r)\right] + \mathbb{E}_{\hat{c},r}\left[\log\left(1 - D(\hat{Z}_o^c, Z_g^r)\right)\right].$$

where $D(a, b) = a^\top W b$ is a bilinear discriminator, and $\hat{r}, \hat{c}$ are negative samples generated by randomly removing a portion of bits (e.g., 20%) from the Bloom filters before encoding them with the dense codebook in Eq 2. This design enhances the fine-grained text terms encoded by Bloom filters. By maximizing the mutual information between regions and their context while discriminating negative inputs with any missing bits, we extract critical geographic associations from shared bits.

## 4.4 Training Strategy and Optimizations

The shared codebook must balance coarse region-level features (for prediction) and fine query-object matches (for retrieval). Direct joint training causes codebook overfitting to retrieval data due to scale disparity: prediction tasks generally uses only thousands of regions, while retrieval tasks can involve millions of user queries. To resolve this, we employs two-phase training: (1) Warm-up Phase: Train exclusively on prediction tasks for some (i.e., 2-3) epochs, and (2) Alternating Phase: Iteratively training on prediction and retrieval data batches. This effectively balances holistic region features while absorbing object specifics (full algorithm in Appendix B).

In addition, we propose to accelerate the computation leveraging Bloom filter sparsity. As shown in Figure 2, we propose row and column selection in the codebook and the evaluation layer, significantly accelerating training and inference computations.

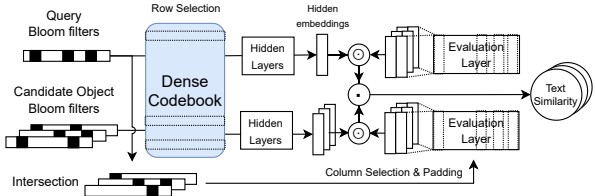

Figure 2: Illustration of the proposed optimization technique.

## 5 Experiments

In this section, we evaluate the output representations on geographic prediction and retrieval tasks following previous literature [1, 39]. We also perform efficiency and ablation studies.

### 5.1 Experimental setups

**Datasets** We use data from two cities, i.e. *Beijing* and *Shanghai*. The datasets include Point-of-Interests (POIs) from Meituan [39], a leading consumer service platform in China. The statistics of the datasets are shown in Table 1.

Table 1: Dataset Statistics

| City | POIs | Labeled Queries | Regions |
|------|------|-----------------|---------|
| Beijing | 122420 | 168,998 | 1010 |
| Shanghai | 116859 | 127,183 | 1358 |

**Downstream tasks and Evaluation Protocols** We evaluate the learned representations on three downstream tasks: *POI retrieval*, *Population Density Prediction*, and *House Price Prediction*. POI retrieval is one of the most common tasks in the location-based service. We evaluate following [39], where the user-selected objects on the Meituan platform are labeled as ground truth and the results are measured with Recall and Normalized Discounted Cumulative Gain (NDCG). Population density and house price prediction are two common tasks in the literature [9], and we measure the results with Mean Absolute Error (MAE), Root Mean Squared Error (RMSE), and Coefficient of Determination ($R^2$). More details for downstream tasks and metrics are provided in Appendix D.

**Baselines** The proposed method is compared with strong GIR and Urban Region Representation Learning baselines, including GraphSAGE [19], DGI [59], MVGRL [20], SpaBERT [35], HGI [24], and CityFM [1] for prediction tasks, and BM25 [53], BERT [13], OpenAI[3], DRMM [18], DrW [39], MGeo [14], and DPR [29] for retrieval tasks. Many other recent prediction methods [28, 34, 62, 69] rely on other data types (e.g., GPS trajectory data of vehicles) as inputs. However, GPS trajectory data are only available in very few cities, and we did not find them for our datasets. Thus, we cannot run these methods for comparison in our experiments. It's noteworthy that we omit PLM-based retrieval methods with sparse representations [4, 15] due to their common failure on common GIR addresses and numbers. More details of these baselines can be found in Appendix E.

### 5.2 Experimental Results in Prediction Tasks

We evaluate the effectiveness of UrbanSparse on Population Density Prediction (population per square kilometer) and on House Price Prediction (CNY per square meter) as in Table 2 and 3. The results give us important insights: (1) Graph contrastive learning methods (DGI, MVGRL), while incorporating geospatial proximity within region graphs, perform similar or worse than a direct

---

[3]https://platform.openai.com/docs/guides/embeddings

feature reconstruction over input features (i.e., GraphSAGE) due to the loss of fine-grained textual features. (2) Competitive baselines HGI and CityFM have mitigated the problem via rule-based geographic context learning, where CityFM pre-trains on extensive OpenStreetMap data to achieve the second-best performance. (3) UrbanSparse stands out by leveraging Bloom filters as fine-grained text features with its novel contrastive learning at multiple granularities. Particularly, we compare it with its variant *UrbanSparse w/o Individual* (where labels from retrieval tasks are removed). The results show that improvements from labeled queries are small (but statistically significant, i.e., T-test p-value $< 0.05$). Without these labels, our framework still outperform other baselines.

Table 2: Population Density Prediction, with the best in **bold** and the second best underlined

| Method | Beijing | | | Shanghai | | |
|---|---|---|---|---|---|---|
| | MAE↓ | RMSE↓ | $R^2$↑ | MAE↓ | RMSE↓ | $R^2$↑ |
| GraphSage | $4566 \pm 361$ | $7113 \pm 811$ | $0.60 \pm 0.06$ | $11020 \pm 807$ | $15142 \pm 1827$ | $0.34 \pm 0.05$ |
| DGI | $5703 \pm 259$ | $8403 \pm 234$ | $0.47 \pm 0.04$ | $13022 \pm 567$ | $17482 \pm 371$ | $0.17 \pm 0.04$ |
| MVGRL | $5675 \pm 248$ | $8397 \pm 378$ | $0.47 \pm 0.05$ | $12266 \pm 633$ | $16739 \pm 774$ | $0.24 \pm 0.01$ |
| SpaBERT | $6494 \pm 432$ | $9088 \pm 856$ | $0.34 \pm 0.06$ | $11586 \pm 672$ | $15401 \pm 1333$ | $0.31 \pm 0.07$ |
| HGI | $4547 \pm 349$ | $7210 \pm 754$ | $0.59 \pm 0.05$ | $8942 \pm 755$ | $13606 \pm 1512$ | $0.46 \pm 0.09$ |
| CityFM | 4420 $\pm 348$ | 6496 $\pm 694$ | 0.66 $\pm 0.04$ | 6930 $\pm 633$ | 10751 $\pm 1184$ | 0.67 $\pm 0.05$ |
| UrbanSparse | **3307** $\pm 14$ | **5772** $\pm 31$ | **0.75** $\pm 0.003$ | **5343** $\pm 55$ | **8958** $\pm 169$ | **0.78** $\pm 0.01$ |
| - w/o Individual | $3368 \pm 53$ | $5871 \pm 78$ | $0.74 \pm 0.01$ | $5467 \pm 81$ | $9189 \pm 177$ | $0.76 \pm 0.01$ |
| (Gains) | 25.16% | 11.14% | 13.64% | 22.90% | 16.68% | 16.42% |

Table 3: House Price Prediction, with the best in **bold** and the second best underlined

| Method | Beijing | | | Shanghai | | |
|---|---|---|---|---|---|---|
| | MAE↓ | RMSE↓ | $R^2$↑ | MAE↓ | RMSE↓ | $R^2$↑ |
| SpaBERT | $21015 \pm 1837$ | $28083 \pm 2289$ | $0.55 \pm 0.05$ | $16772 \pm 1253$ | $25340 \pm 3405$ | $0.31 \pm 0.08$ |
| GraphSage | 14239 $\pm 1824$ | 19947 $\pm 2836$ | 0.77 $\pm 0.05$ | $17408 \pm 1065$ | $25118 \pm 3217$ | $0.32 \pm 0.05$ |
| DGI | $16203 \pm 1352$ | $22399 \pm 1796$ | $0.70 \pm 0.05$ | $18415 \pm 426$ | $25876 \pm 600$ | $0.14 \pm 0.04$ |
| MVGRL | $16799 \pm 2035$ | $23792 \pm 2753$ | $0.66 \pm 0.09$ | $17833 \pm 211$ | $25011 \pm 237$ | $0.20 \pm 0.01$ |
| HGI | $14974 \pm 1251$ | $21833 \pm 2022$ | $0.72 \pm 0.06$ | $16095 \pm 1140$ | $24022 \pm 3478$ | $0.38 \pm 0.06$ |
| CityFM | $17721 \pm 2178$ | $24123 \pm 2884$ | $0.66 \pm 0.06$ | 15694 $\pm 1727$ | $24862 \pm 4444$ | $0.33 \pm 0.15$ |
| UrbanSparse | **11983** $\pm 507$ | **17155** $\pm 767$ | **0.82** $\pm 0.01$ | **13281** $\pm 325$ | **20610** $\pm 671$ | **0.46** $\pm 0.04$ |
| - w/o Individual | $12881 \pm 569$ | $18326 \pm 691$ | $0.80 \pm 0.02$ | $13756 \pm 166$ | $21669 \pm 535$ | $0.40 \pm 0.03$ |
| (Gains) | 15.84% | 14.00% | 6.49% | 15.38% | 13.90% | 17.95% |

## 5.3 Experimental Results in Retrieval Tasks

We evaluate UrbanSparse in POI retrieval tasks against strong retrieval baselines, where standard deviations are omitted as they are very small ($< 0.003$). As the vanilla BM25, BERT, OpenAI, and DPR only consider text similarity, we supplement BM25-D, BERT-D, OpenAI-D, and DPR-D to incorporate geographic distances following [39] by defining $Relevance(q, o) = (1 - \alpha)(1 - D_{norm}(q, o)) + \alpha \cdot T_{norm}(q, o)$, where $D_{norm}(q, o)$ denotes the geographic distances, $T_{norm}$ denotes the text similarity from the vanilla baseline, both are normalized to $[0, 1]$. $\alpha$ is a hyper-parameter balancing the text and the distance similarities, set by grid searching on the dev set. In addition, DRMM, DrW, DPR, UrbanSparse are fine-tuned on labeled queries while the rest are not. The results in Table 4 provide several key insights: (1) The classical term-matching method BM25-D significantly outperform vector-based methods BERT-D and the leading commercial product, OpenAI-D. This underscores the critical importance of term-matching capabilities in retrieval tasks. (2) UrbanSparse surpasses heavy-weight BERT-based methods such as DrW and DPR-D, showcasing the effectiveness of evaluating Bloom filter bits. Furthermore, its superiority over its variant without the Collective View (denoted as w/o Collective) validates the benefits of incorporating prediction tasks.

## 5.4 Efficiency Studies

**Training Time** We evaluate the training time of UrbanSparse against top-performing baselines on 1 NVIDIA V100 32GB. As shown in Table 5, DPR, GraphSAGE, HGI, CityFM, and DrW require

Table 4: Point-of-Interest Retrieval, with the best in **bold** and the second best underlined

| Method | Beijing | | | Shanghai | | |
|--------|---------|---------|---------|----------|---------|---------|
| | Recall@10 | NDCG@5 | NDCG@1 | Recall@10 | NDCG@5 | NDCG@1 |
| BM25 | 0.3401 | 0.2199 | 0.1634 | 0.3274 | 0.1913 | 0.1260 |
| BM25-D | 0.5477 | 0.4263 | 0.3569 | 0.6484 | 0.5215 | 0.4380 |
| BERT | 0.1602 | 0.1169 | 0.0979 | 0.1277 | 0.0853 | 0.0662 |
| BERT-D | 0.2400 | 0.1614 | 0.1298 | 0.2622 | 0.1687 | 0.1233 |
| OpenAI | 0.3265 | 0.2157 | 0.1637 | 0.3213 | 0.1875 | 0.1258 |
| OpenAI-D | 0.5206 | 0.3803 | 0.3078 | 0.6313 | 0.4852 | 0.3864 |
| DRMM | 0.1773 | 0.1105 | 0.0758 | 0.1921 | 0.1287 | 0.0804 |
| DRMM-D | 0.4357 | 0.2378 | 0.1566 | 0.4380 | 0.2433 | 0.1595 |
| DrW | 0.6316 | 0.4814 | 0.3791 | 0.7159 | 0.5394 | 0.4114 |
| DPR | 0.4183 | 0.2775 | 0.2121 | 0.4087 | 0.2498 | 0.1746 |
| DPR-D | 0.6688 | 0.4980 | 0.4132 | 0.7281 | 0.5641 | 0.4554 |
| UrbanSparse | **0.7062** | **0.5734** | 0.4990 | **0.7589** | **0.6209** | **0.5315** |
| - w/o Collective | 0.6988 | 0.5695 | **0.4991** | 0.7526 | 0.6157 | 0.5289 |
| (Gains) | 5.59% | 15.14% | 20.76% | 4.23% | 10.07% | 16.71% |

Table 5: Training Time (Minutes) and Inference Memory (MB)

(a) Prediction Tasks

| Method | Training Time | | Memory Usage | |
|--------|---------|----------|---------|----------|
| | Beijing | Shanghai | Beijing | Shanghai |
| GraphSAGE | 24 | 19 | 3.95 | 3.95 |
| HGI | 281 | 510 | 0.25 | 0.33 |
| CityFM | 510 | 355 | 3.95 | 3.95 |
| UrbanSparse | 22 | 11 | 0.25 | 0.33 |
| (Saves) | 8.33% | 42.11% | 0.00% | 0.00% |

(b) Retrieval Tasks

| Method | Training Time | | Memory Usage | |
|--------|---------|----------|---------|----------|
| | Beijing | Shanghai | Beijing | Shanghai |
| OpenAI | N/A | N/A | 717.30 | 684.72 |
| DrW | 142 | 97 | 11806.29 | 10210.42 |
| DPR-D | 448 | 282 | 507.80 | 491.60 |
| UrbanSparse | 51 | 33 | 71.92 | 66.40 |
| (Saves) | 64.08% | 65.97% | 85.84% | 86.49% |

considerably longer training time. This is particularly evident for DPR and CityFM as they rely on fine-tuning BERT, resulting in substantial training overhead. UrbanSparse leverages the sparsity of Bloom filters to significantly reduce computational demands.

**Inference Memory Usage**   We also report the memory usage in inference, with all embeddings stored in 32-bit float numbers. For prediction tasks, as the trained models can be offloaded and urban regions are relatively few, the memory usage across methods exhibits negligible differences. In retrieval tasks, however, significant differences emerge due to the model parameters needed to process user queries and a large amount of POIs. UrbanSparse produces sparse representation with a fixed dimension of 8192 and a density of only 2–3%, achieving dramatically reduced memory usage, making it a resource-efficient choice for retrieval tasks.

**Query Processing Speed**   We further evaluate the query processing speed of UrbanSparse against DPR-D by running a brute-force search. We do not evaluate DrW as it requires $> 24$ hours for evaluation. As shown in Table 6, UrbanSparse significantly outperforms DPR-D, achieving approximately 3.6x and 3.8x higher Queries Per Second (QPS) in Beijing and Shanghai, respectively. This substantial improvement is attributed to UrbanSparse's small model size and high representation sparsity.[4]

Table 6: Query Per Second Comparison

| Method | #Params (M) | Beijing | Shanghai |
|--------|-------------|---------|----------|
| DPR-D | 110 | 133.05 | 133.94 |
| UrbanSparse | 2.72 | 476.29 | 505.20 |

---

[4]While we achieve a $40\times$ reduction in parameters, the $4\times$ QPS gain is bounded by our custom CUDA kernels for sparse representation calculations: the non-coalesced memory accesses in our kernel and an insufficiently optimized kernel dispatch strategy incur significantly higher latency than vendor-optimized dense kernels from experts. We anticipate that expert-tuned kernels will further narrow this gap.

**Scalability** UrbanSparse is lightweight and can theoretically scale up due to low computational complexity. To empirically verify this, we evaluate on established large datasets GeoGLUE [31], a public GIR benchmark with 2,849,754 POIs. However, the benchmark doesn't support prediction tasks as it contains over 50% fake POIs for anonymity. As shown in Table 7, our methods show competitive retrieval effectiveness while trains much faster than PLM-based method DPR-D. DrW and DRMM are omitted as they reports OOM on this dataset.

Table 7: Point-of-Interest Retrieval and Training Time on GeoGLUE

| Method | Recall@10 | NDCG@5 | NDCG@1 | Training Time (Min) |
|---|---|---|---|---|
| MGeo [14] | N/A | N/A | 0.5270 | Unknown |
| DPR-D | 0.7611 | 0.6318 | **0.5350** | 131 |
| UrbanSparse | **0.7621** | **0.6344** | 0.5310 | **54** |

## 5.5 Ablation Studies

While UrbanSparse's prediction advantages stem from Bloom filter-contrastive learning integration, can other contrastive learning methods replicate this by simply adopting Bloom filters? Our evaluation shows fundamental limitations: Table 8 reveals Bloom filters' inconsistent impact. While producing 150%+ improvements for Shanghai population prediction with DGI/MVGRL, other tasks show mixed results (-11.4% to +36.3%). In conclusion, standard contrastive objectives appear poorly suited to extract Bloom filters' encoded bit patterns, while UrbanSparse achieves consistent gains. We also studied other hyperparameters (e.g., tokenizers, hash functions, Bloom filter length), and put these technical details in Appendix F.

Table 8: Effect of Bloom Filters vs BERT Embeddings

| Method | Pop. Pred. $R^2\uparrow$ | | House Pred. $R^2\uparrow$ | |
|---|---|---|---|---|
| | Beijing | Shanghai | Beijing | Shanghai |
| DGI | 0.4682 | 0.1877 | 0.7240 | 0.1511 |
| w/ Bloom filters | 0.4807 | 0.4752 | 0.7521 | 0.2060 |
| (Gains) | 2.7% | 153.2% | 3.9% | 36.3% |
| MVGRL | 0.4483 | 0.1716 | 0.6686 | 0.1866 |
| w/ Bloom filters | 0.3974 | 0.4416 | 0.7085 | 0.2130 |
| (Gains) | -11.4% | 157.2% | 6.0% | 14.2% |
| UrbanSparse$_{(BERT)}$ | 0.3830 | 0.3046 | 0.3388 | 0.1010 |
| UrbanSparse | 0.7480 | 0.7805 | 0.8234 | 0.4612 |
| (Gains) | 95.3% | 156.2% | 143.0% | 356.6% |

## 6 Conclusion

In this work, we introduced UrbanSparse, a unified framework integrating the traditionally separate geospatial prediction and retrieval tasks via a two-view learning mechanism that combines Bloom filter-based sparse representations and graph contrastive learning. Extensive experiments demonstrate its ability to outperform state-of-the-art baselines while achieving significant gains in efficiency, accuracy, and scalability. This study pioneers a new research direction in urban computing, offering a transformative solution to unify task frameworks and address complex urban challenges with greater resource efficiency.

## Acknowledgment

This research/project is supported by the National Research Foundation, Singapore, under its AI Singapore Programme (AISG Award No: AISG2-PhD-2021-08-020[T]) and A*STAR RIE2025 Manufacturing, Trade and Connectivity (MTC) Programmatic Fund (M24N6b0043) administered by A*STAR. Yuanlong Chen's work is supported by the Singapore International Graduate Award.

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

## A   Bloom Filters and Membership Tests

A Bloom filter is a space-efficient probabilistic data structure designed for set membership testing. Given a set $A = \{a_1, a_2, \ldots, a_n\}$ of $n$ elements, a Bloom filter encodes $A$ using a bit vector $B$ of length $m$, initially filled with zeros. It relies on $k$ independent hash functions $H_1, H_2, \ldots, H_k$, each mapping an input element to a position in $\{1, \ldots, m\}$.

To insert an element $a \in A$, the bits at indices $H_1(a), H_2(a), \ldots, H_k(a)$ in the bit vector $B$ are set to 1. To check whether a query element $q$ belongs to the set, the Bloom filter examines the bits at positions $H_1(q), H_2(q), \ldots, H_k(q)$. If any of these bits is 0, $q$ is definitely not in $A$. If all are 1, the Bloom filter reports that $q$ may belong to $A$, introducing a false positive probability but guaranteeing no false negatives.

This tradeoff makes Bloom filters particularly useful in large-scale applications where space efficiency and fast membership queries are critical. In our implementation, we empirically found $m \geq 8192$, $k \geq 2$ sufficient for small false positive rates (See table 11). We use SHA-256 as random hash functions, leaving more sophisticated designs to future work.

## B   Training Algorithm

We hereby detail the training algorithm used to balance the training on retrieval and prediction tasks in our framework.

---

**Algorithm 1** Two-Phase Training of UrbanSparse

---

1: **Input**: $\mathcal{D}_{\text{pred}}$ (region data), $\mathcal{D}_{\text{retr}}$ (query-object pairs), $E_{\text{warm}} = 3$, $E_{\text{total}} = 20$
2: **Parameter**: Model $f_\theta$ with codebook $\mathbf{C}$

3: **procedure** WARM-UP PHASE
4:     **for** epoch $= 1$ to $E_{\text{warm}}$ **do**
5:         **for** each batch $B \in \mathcal{D}_{\text{pred}}$ **do**
6:             Compute $\mathcal{L}_{\text{pred}}$
7:             Update $\theta \leftarrow \theta - \eta \nabla_\theta \mathcal{L}_{\text{pred}}$
8:         **end for**
9:     **end for**
10: **end procedure**

11: **procedure** ALTERNATING PHASE
12:     **for** epoch $= E_{\text{warm}} + 1$ to $E_{\text{total}}$ **do**
13:         Shuffle $\mathcal{D}_{\text{pred}}$ and $\mathcal{D}_{\text{retr}}$
14:         **for** $i = 1$ to $\max(|\mathcal{D}_{\text{pred}}|, |\mathcal{D}_{\text{retr}}|)$ **do**
15:             Sample batch $B_p \sim \mathcal{D}_{\text{pred}}$, $B_r \sim \mathcal{D}_{\text{retr}}$
16:             Compute $\mathcal{L}_{\text{pred}}$ on $B_p$ via Eq.3
17:             Compute $\mathcal{L}_{\text{retr}}$ on $B_r$ via LambdaRank
18:             Update $\theta \leftarrow \theta - \eta \nabla_\theta (\mathcal{L}_{\text{pred}} + \mathcal{L}_{\text{retr}})$
19:         **end for**
20:     **end for**
21: **end procedure**

---

## C   Complexity Analysis

We analyze the time complexity of the proposed UrbanSparse in retrieval tasks. The prediction tasks are not analyzed as they vary significantly with downstream predictors. Let $h_i$ be the output dimension of the i-th model layer, $u$, $e$ the count of non-zero bits in the user query and object Bloom filters, the time complexity of model forward process during training is given by $O((u + e)h_1 + h_1 h_2 + h_2 h_3 + min(u, e)h_3)$. As $h_1$, $h_2$, and $h_3$ are small constants, (i.e., 256 and 32 in our implementation), the training time is dominated by $u + e$, which is only affected by the query and object text lengths, and the number of hash functions. The time complexity during inference is $O(u(h_1 + h_3) + h_1 h_2 + h_2 h_3)$, which is dominated by the query length.

## D Downstream Tasks and Evaluation Protocols

To evaluate the quality of the learned representations, we consider three downstream tasks:

- **POI Retrieval.** Given a user query, retrieve relevant points of interest (POIs). We follow the dataset split and protocol of [39], where Meituan user-selected POIs serve as ground truth.
- **Population Density Prediction.** Predict the population density of a geographic region based on its learned embedding.
- **House Price Prediction.** Estimate the average house price in a region using its representation.

It is noteworthy that land use, population density, and house price prediction are the top-3 common tasks according to the recent survey [9]. However, we fail to find high-quality land use ground truth in the two studied cities, so we only evaluate the latter two tasks.

**Evaluation Metrics**   We employ the following metrics for each task:

- *Recall@K* and *NDCG@K* (Normalized Discounted Cumulative Gain) for POI retrieval. Recall@$K$ measures the fraction of ground-truth POIs appearing in the top-$K$ results, while NDCG@K accounts for both relevance and ranking position.
- *Mean Absolute Error (MAE)*, *Root Mean Squared Error (RMSE)*, and *Coefficient of Determination ($R^2$)* for the regression tasks (population density and house price). MAE and RMSE quantify absolute and squared deviations, respectively; $R^2$ indicates the proportion of variance explained by the model.

For retrieval tasks, we requested and got the established benchmark from [39], which has a fixed split with the train/dev/val ratio 0.81:0.09:0.10. As the splits are fixed without randomness, the standard deviations appear to be very small ($< 0.003$ for all methods) and we omit the standard deviations in our table. For prediction tasks, we follow common practice of unsupervised representation learning, evaluating the learned representations with scikit-learn RandomForestRegressor on all urban regions using 5-fold cross-validation. We strictly repeat all experiments 10 times, report the average results and standard deviations without cherry-picking.

**Data Sources**   All datasets (or their corresponding embeddings/Bloom filters) used in this paper are publicly available. Table 9 lists each data type along with its source and download link.

Table 9: Data sources and download links

| Data Type | Source | Link |
|---|---|---|
| POI datasets and queries | Meituan | `https://anonymous.4open.science/r/UrbanSparse` |
| Population density | WorldPop | `https://hub.worldpop.org` |
| House prices | Beike | `https://ke.com` |
| Administrative boundaries | GADM | `https://gadm.org` |

Due to licensing constraints, the raw Meituan query and POI text data cannot be shared. Instead, we provide the corresponding Bloom filters and geographic coordinates in our GitHub repository, along with BERT, OpenAI, and our trained DPR embeddings to enable full replication of all experiments.

## E Baselines and Implementation Details

We compared with the following baselines from the prediction and retrieval tasks respectively:

(1) **Prediction Methods**

- GraphSage [19]: This classical graph learning algorithm samples and aggregates neighbor nodes to compute node embeddings. It is commonly used as a geospatial representation learning baseline with node feature or graph structure reconstruction objectives.It is noteworthy that we have tested both vanilla GCN and GraphSAGE as representative graph learning baselines. However, GCN suffers from scalability issue and reports OOM in our datasets.

- DGI [59]: This method maximizes the mutual information between node and graph embeddings. We take its graph embedding as the region representation. It doesn't explicitly learn geospatial correlations.
- MVGRL [20]: Inspired by DGI, this method maximizes the mutual information between the node and graph embedding from the original graph and an augmented graph constructed by graph diffusion. We use its graph embedding as the region representation. It doesn't explicitly learn geospatial correlations.
- SpaBERT [35]: This method utilizes pre-trained BERT to learn geographic object representations with text and geospatial proximity. We average its object embeddings as region representation.
- HGI [24]: Inspired by DGI, this method incorporates geospatial domain knowledge by hierarchically maximizing the mutual information between POI, region, and city representations. It proposes a novel rule-based strategy of positive and negative sampling to preserve fine-grained and holistic information simultaneously.
- CityFM [1]: This method learns general-purpose geospatial representations from multimodal OpenStreetMap node, polyline, and polygon data. We use its node encoder to encode POI representations and average them as the region representation.

(2) **Retrieval Methods**
- BM25 [53]: This classical information retrieval method computes text similarities based on bag-of-words (BOW) representations and term-matching.
- BERT [13]: BERT is a representative pre-trained language model that excels in capturing deep semantics. We use the cosine similarity between queries and object representations to assess text similarities.
- OpenAI [5]: OpenAI's text-embedding-3-small generates high-quality text embedding effective for retrieval tasks. Its technical details remain proprietary.
- DRMM [18]: This model evaluates text similarities based on pairwise local interactions at the term level. It doesn't account for geographic proximity.
- DrW [39]: This method utilizes BERT to capture term-level text similarities and propose a novel query-aware combination strategy with geospatial distances.
- DPR [29]: This method finetunes BERT on labeled queries, shortening the distance between textual similarities between relevant query-object pairs.
- MGeo [14]: This method applies multi-task pre-training on a BERT-based encoder and finetunes it by user queries. As we are unable to replicate the results of MGeo with their official code, we don't evaluate on the two datasets in the main content of the paper, and only reference the evaluation results on GeoGLUE in Table 7 as presented by the authors.

(3) **UrbanSparse Variants**
- UrbanSparse w/o Individual, where we remove the proposed Individual View in Figure 1.
- UrbanSparse w/o Collective, where we remove the proposed Collective View in Figure 1.

It is worth noting that although many recent methods for urban region representation learning rely on human mobility data (e.g., vehicle trajectories), such data are available for only a limited number of cities, so we do not include them in our comparisons. Instead, to ensure relevance, we compare against the most recent versions of HGI (2023) and CityFM (2024). On the other hand, sparse retrieval methods such as SPLADE [15] and BGE-M3 [4] rely on PLM tokenizers that split each digit of an address number into a separate token. As a result, they cannot properly match street or house numbers and cannot work properly in our datasets.

For retrieval baselines BM25, BERT, OpenAI, and DPR only consider text similarity, we supplement BM25-D, BERT-D, OpenAI-D, and DPR-D to incorporate geographic distances following [39] by defining $Relevance(q, o) = (1 - \alpha)(1 - D_{norm}(q, o)) + \alpha \cdot T_{norm}(q, o)$, where $D_{norm}(q, o)$ denotes the geographic distances, $T_{norm}$ denotes the text similarity from the vanilla baseline, both are normalized to $[0, 1]$. $\alpha$ is a hyper-parameter balancing the text and the distance similarities, set by grid searching on the dev dataset as in Table 10.

The representation dimension $d$ varies among different baselines. We set $d = 64$ for HGI, $d = 512$ for DGI and MVGRL, $d = 768$ for BERT SpaBERT, and DPR, $d = 1024$ for CityFM and GraphSAGE,

---

[5]https://platform.openai.com/docs/guides/embeddings

Table 10: $\alpha$ value for baselines

| Method | Beijing | Shanghai |
|--------|---------|----------|
| BM25-D | 0.4 | 0.4 |
| BERT-D | 0.4 | 0.4 |
| OpenAI-D | 0.3 | 0.3 |
| DRMM-D | 0.7 | 0.7 |
| DPR-D | 0.3 | 0.3 |

and $d = 1536$ for OpenAI, following the settings recommended in the corresponding paper. For the proposed UrbanSparse, we fix the Bloom filter length to $m = 8192$ with $k = 2$ SHA-256 hash functions. In prediction tasks, we set the output region representation dimension $d = 64$. For retrieval tasks, all methods run a brute-force search over all POIs unless otherwise specified. All experiments are conducted on 1 NVIDIA V100 32 GB.

# F    Additional Ablation Studies

## F.1    Bloom Filter Length & Number of Hash functions

We analyze the effect of Bloom filter length $m$ and the number of hash functions $k$ using NDCG@5 on the POI retrieval in Beijing, chosen for its low standard deviation (below 0.002) and sensitivity to Bloom filter changes. As shown in Table 11, $m < 2048$ and $k < 2$ lead to worse performance due to insufficient capacity, while $m > 8192$ or $k > 2$ yields negligible gains, suggesting that Bloom filters reach their optimal capacity when $m$ and $k$ are sufficient to encode the geographic vocabulary, and further increases offer no additional benefits.

Table 11: Effect of $k$ and $m$ on Beijing POI Retrieval

| $k$ \ $m$ | 512 | 2048 | 8192 | 32768 |
|-----------|-----|------|------|-------|
| 1 | 0.5392 | 0.5382 | 0.5569 | 0.5464 |
| 2 | 0.5578 | 0.5689 | 0.5724 | 0.5738 |
| 3 | 0.5530 | 0.5689 | 0.5717 | 0.5730 |
| 8 | 0.5312 | 0.5679 | 0.5717 | 0.5727 |

## F.2    Effect of Tokenizers

We analyze how the choice of tokenizers affects the retrieval effectiveness of UrbanSparse. We tested n-gram tokenizers as in DSSM [55] and a dictionary-based tokenizer Jieba (`https://github.com/fxsjy/jieba`). Table 12 shows that the combination of 1-gram, 2-gram, and dictionary-based tokenizers achieves the best Recall@20 and NDCG@5, enhancing the term-matching capability of Bloom filters.

Table 12: Effect of Tokenizers on Beijing POI retrieval

| Tokenizer | Recall@20 | NDCG@5 |
|-----------|-----------|--------|
| 1-gram | 0.7022 | 0.5547 |
| 2-gram | 0.7175 | 0.5623 |
| 3-gram | 0.6642 | 0.4511 |
| 1,2,3-gram | 0.7133 | 0.5551 |
| Dict. (Jieba) | 0.7264 | 0.5633 |
| 1,2-gram+Dict. | **0.7427** | **0.5740** |

## F.3    Effect of Context Graph Construction

The context graph, constructed by randomly sampling from the $K$-hop neighbors of objects within the region graph, plays a critical role in the proposed Collective View. Larger $K$ values create more

diverse and comprehensive context graphs, which can enhance the model's ability to capture complex relationships. As shown in Table 13, $K = 3$ and $K = 4$ achieve the highest effectiveness for both population and house price prediction tasks. $K > 4$ yields no significant gains in effectiveness.

Table 13: Effect of $K$-hop Context Graphs

| K | Pop. Pred. $R^2$↑ | | House Pred. $R^2$↑ | |
|---|---|---|---|---|
| | Beijing | Shanghai | Beijing | Shanghai |
| 1 | 0.6035 | 0.6614 | 0.7656 | 0.3751 |
| 2 | 0.6818 | 0.7228 | 0.7910 | 0.4470 |
| 3 | 0.7399 | 0.7857 | 0.8200 | 0.4530 |
| 4 | 0.7480 | 0.7805 | 0.8234 | 0.4612 |

### F.4  Effect of Training Algorithms

We evaluate the effect of the proposed training algorithm as in Algorithm 1 by (1) training on two datasets separately in each epoch instead of interweaving each data batch and (2) removing the warm-up epochs. The results in Table 14 demonstrate that these modifications lead to reduced effectiveness in either the prediction or retrieval task. This suggests that the proposed training algorithm successfully trained a codebook to share useful information between the two tasks.

Table 14: Effect of Training Algorithms

| Method | Pop. Pred. $R^2$↑ | | Retrieval NDCG@5↑ | |
|---|---|---|---|---|
| | Beijing | Shanghai | Beijing | Shanghai |
| UrbanSparse | 0.7480 | 0.7805 | 0.5734 | 0.6209 |
| Train separately | 0.7001 | 0.7450 | 0.5459 | 0.6140 |
| No warm-up | 0.7290 | 0.7670 | 0.5553 | 0.6162 |

### F.5  Effect of Row/Column Selection

We evaluate the efficiency gain of the two sparsification optimizations described in Figure 2, i.e., *row selection* and *column selection*, by analyzing the training time (minutes) and training memory usage (MB) on both cities. As shown in Table 15, both optimizations reduce memory footprint via sparsifying dense computations. *Column selection* alone yields $> 4\times$ speedup ($214 \rightarrow 51$ min) and $> 2\times$ memory reduction ($166.5 \rightarrow 71.9$ MB) by computing only query–bit–matched entries. *Row selection* brings a smaller but complementary gain ($72 \rightarrow 51$ min). When POI Bloom filters contain many bits, the sparse–dense kernel in PyTorch does not significantly outperform dense multiplication due to memory access patterns, so column selection delivers the major efficiency gains while row selection acts as a secondary refinement.

Table 15: Effect of Row and Column Selection Optimizations on Training Efficiency.

| Method | Training Time (min)↓ | | Training Memory (MB)↓ | |
|---|---|---|---|---|
| | Beijing | Shanghai | Beijing | Shanghai |
| UrbanSparse | 51 | 33 | 71.92 | 66.40 |
| w/o Row Selection | 72 | 40 | 127.38 | 112.21 |
| w/o Column Selection | 214 | 133 | 166.51 | 147.75 |

## G  Further Discussions

### G.1  Limitations and Future Work

First, UrbanSparse unifies prediction and retrieval through Bloom-filter encodings and learned embeddings, but it presumes sufficiently rich geo-textual inputs and discards text sequence information,

which may degrade performance on sparse or highly noisy data. In future work, two straightforward strategies may be useful to strengthen robustness: (1) Designing off-the-shelf query rewriting modules to formalize queries and eliminate noises. (2) Explicitly marking empty areas with special markers (e.g., random points) to better inform the model [33]. Second, all our experiments rely on Meituan data from Beijing and Shanghai, and the model's hyperparameters (e.g. filter size, hash count, training schedule) were tuned for these cities, potentially limiting generalization to other urban environments. Third, the prediction tasks in this paper only involve the prediction from known areas to known areas. Future work should consider the spatio-temporal prediction tasks [46, 47], such as time series forecasting [36, 37], imputation [41], and recovery [61]. Finally, UrbanSparse may inadvertently reflect or amplify existing biases in spatial data sources (i.e., POIs in this work), thereby reinforcing socioeconomic disparities across urban regions. Future work should adopt fairness-aware sampling and noise-injected query augmentation during training, and include systematic bias auditing and fairness calibration across cities and demographic groups.

## G.2 Broader Impact

By improving population density and house-price estimates and enhancing POI retrieval, UrbanSparse can aid urban planning, resource allocation, and user-facing location services while reducing computational costs. However, the use of fine-grained user queries and POI data risk privacy concerns, models trained on major-city data may underperform in underserved regions, and high-precision retrieval could be misused for targeted marketing or surveillance.

