# OpenReview forum: "Reconciling Geospatial Prediction and Retrieval via Sparse Representations"
_NeurIPS.cc/2025/Conference — NeurIPS 2025 poster_

### Official Review · Reviewer_vg91 · 2025-06-24

**Clarity:** 3
**Significance:** 3
**Originality:** 3
**Rating:** 4
**Confidence:** 4

**Summary:**

The paper is related to urban computing. It proposes UrbanSparse, a unified framework addressing the novel challenge of integrating geospatial prediction and retrieval tasks. The system encodes fine-grained text details for retrieval and preserves features for prediction via Bloom filters and a two-view learning mechanism. An extensive experimental evaluation on two real-world datasets shows that UrbanSparse significantly improves performance over existing methods, while achieving significant gains in efficiency, accuracy, and scalability.

**Questions:**

Please refer to the weaknesses.

**Ethical Concerns:**

["NO or VERY MINOR ethics concerns only"]

**Limitations:**

Yes

**Quality:**

3

**Strengths And Weaknesses:**

Strengths

S1. The paper tackles a long standing challenge for the urban computing: unifying geospatial prediction and retrieval under a single representation learning framework, which could be essential for future foundation models.

S2. The motivations are clear. Urban computing relies on massive city‐scale data, but existing approaches train separate models for tasks like socioeconomic inference and geographic searches. Combining them into one embedding can save labor and boost efficiency.

S3. The experiments are extensive and detailed. They use two real‐world urban datasets, evaluate both accuracy and runtime, and compare against strong baselines in prediction and retrieval; the reported improvements are meaningful.

S4. The paper is clearly written and easy to follow.

Weakness:

W1. While I find the motivation to be novel, an intuitive running example based on real data inserted in the Introduction may be better for the the reader to understand the importance for the final user to integrate prediction/retrieval urban services.

W2. The chosen datasets (Beijing, Shanghai) reflect large, densely populated cities, so it is unclear if the results extend to cities with different spatial layouts. For example, how would the model perform in cities characterized by sprawling suburban regions? And both datasets contain fewer than 200,000 POIs, which raises the question of whether the approach can handle much higher POI densities. How does varying POI density affect the model’s performance?

W3. While the Bloom filter with two-view contrastive learning empirically works, the paper lacks an intuitive or formal theoretical explanations.

---

> ### Author Rebuttal · Authors · 2025-07-30
>
> Thank you for your careful reading and constructive feedback. We would like to address each point in turn.
>
> **W1. Running Example.**
> We have added a concise running example to the Introduction to show how prediction and retrieval interact in practice: Suppose our system analyzes recent land‑use and economic data across a city and flags a particular neighborhood, home to several hundred restaurants and cafés, as an emerging “culinary hotspot”. When a user searches for “best sushi nearby”, the retrieval module does not simply return the most similar sushi bars city‑wide; it prioritizes those within the hotspot that have seen a surge in visits and searches over the past month. Likewise, a query for “top‑rated vegan restaurants” surfaces vegan venues in the same area that are gaining traction. By combining prediction (detecting dynamic hotspots) with retrieval (serving context‑aware results), the app delivers recommendations that are both timely and closely aligned with the city’s evolving landscape.
>
> **W2. Different spatial layouts and POI density.**
> Handling spatial diversity and POI density is orthogonal to our core contribution, yet we appreciate the your concern and respond as follows:
>
> 1. Our Beijing dataset already spans more than 16 000 km² and includes urban, peri‑urban, and suburban zones, providing naturally diverse patterns. We will add a dedicated case study on a low‑density suburban region of Beijing in the revised Appendix.
> 2. We further evaluate UrbanSparse on the GeoGLUE benchmark (2 849 754 POIs) and show in Appendix F.1 that our model matches strong PLM‑based retrieval baselines (Recall\@10 = 0.760 vs. 0.761 for DPR) while training in 54 minutes, more than twice as fast as DPR.
>
> **W3. Intuitive explanation for the Bloom‑filter contrastive learning.**
> To clarify why our method works, we will insert the following intuitive explanation into Section 4:
>
> 1. *Bit‑level InfoMax.* A discriminator‑based (InfoMax) loss operates on individual Bloom filter bits. Negative POI pairs are discriminated on whether any bits are missing, mimicking the Bloom filter’s membership test, ensuring the detection of important bits.
> 2. Region‑level InfoMax: We group POIs into regions via administrative boundaries, and train a second discriminator to judge whether one region’s aggregated representation is contained within a larger region’s, also analogous to the Bloom filter’s membership test, ensuring spatial coherence across scales.
>
> We hope this intuitive explanation clarifies the roles of the two contrastive objectives and resolves your concern.

---

### Official Review · Reviewer_7EV1 · 2025-06-30

**Clarity:** 3
**Significance:** 4
**Originality:** 3
**Rating:** 5
**Confidence:** 4

**Summary:**

This paper introduces UrbanSparse, an innovative unified architecture for both geospatial prediction and geographic information retrieval. The core innovation lies in leveraging a Bloom filter-based sparse text encoding, integrated with a two-view learning framework that combines fine-grained token-level and regional context signals. It is evaluated on two real-world urban datasets, demonstrating notable gains in accuracy and also improvement in efficiency.

**Questions:**

1. Insufficient Justification for LambdaRank Loss (L209): The choice of LambdaRank introduces computational overhead and complexity, but the paper lacks comparison with simpler alternatives (e.g., static ranking weights or margin-based losses). No ablation is provided to demonstrate its necessity or impact.

2. Unclear Benefit of Row/Column Selection Optimization (L250). The proposed row and column selection technique is claimed to accelerate computation, but the mechanism is not well-explained, nor is its contribution isolated in an ablation or runtime analysis.

3. In Table 6: Despite a 50× reduction in parameter size compared to DPR-D, UrbanSparse achieves only ~4× improvement in QPS. Can the authors comment on the remaining bottlenecks? For instance, does hashing, memory access, or bit-level operations dominate the latency?

**Ethical Concerns:**

["NO or VERY MINOR ethics concerns only"]

**Final Justification:**

Thanks to the authors for the rebuttal content, and my questions are resolved properly. I suggest including these ablation and discussion about performance bottleneck into camera camera-ready version or the appendix.

**Limitations:**

yes

**Paper Formatting Concerns:**

-

**Quality:**

3

**Strengths And Weaknesses:**

S1: This paper is well-written and well-motivated to have a unified architecture to tackle two tasks that benefit each other. It provides a detailed explanation to justify the selection of each module of the framework and provides a clear mathematical formulation.

S2: Figures are helpful and informative to understand the overall architecture.

S3: In terms of significance, this work unifies prediction and retrieval tasks that is timely and practically relevant, especially in this urban computing scenario. It demonstrates substantial improvement in both accuracy and efficiency.

W1: The choice of LambdaRank as the loss function for retrieval is not well justified. The added complexity (e.g., tuning, stability) is not clearly offset by demonstrable gains; no ablation against simpler alternatives is provided.

W2: The benefits of the row/column selection optimization are not sufficiently isolated or quantified. It is unclear how much these contribute to the observed efficiency improvements.

W3: Limited Scalability and Generalization Discussion. The evaluation is limited to two cities, with no evidence or discussion of how well UrbanSparse generalizes to other geospatial contexts, data types (e.g., traffic, satellite), or larger-scale urban systems.

---

> ### Author Rebuttal · Authors · 2025-07-30
>
> Thank you for the careful reading. We would like to address your concern as follows:
>
> **W1 and Q1. Clarification of LambdaRank choice.**
>
> We clarify that loss‑function design is not our primary contribution, and we adopt LambdaRank (NIPS, 2016) because it is a well-known one in learning‑to‑rank practices. Nevertheless, we fully agree that ablation studies on loss functions would help us identify the source of performance gain. Hence, we conduct an ablation study on Beijing datasets. While a static ranking‑weight loss (true=1, false=0) fails to converge, a simple contrastive loss (the same as in DPR) leads to very close results, with 1.2\% performance drop in NDCG@10 and 0.4\% gain in NDCG@1. We are still investigating other loss functions and will provide additional results in the appendix.
>
> **W2 and Q2. Isolation of row/column selection benefits.**
>
> We apologize for the missing ablation studies. We have now conducted ablations on both optimizations:
>
> | Method                               | Training Time (min) |              | Training Memory Usage (MB) |              |
> | ------------------------------------ | ------------------- | ------------ | -------------------------- | ------------ |
> |                                      | **Beijing**         | **Shanghai** | **Beijing**                | **Shanghai** |
> | **UrbanSparse**                      | 51                  | 33           | 71.92                      | 66.40        |
> | **UrbanSparse w/o row selection**    | 72                  | 40           | 127.38                     | 112.21       |
> | **UrbanSparse w/o column selection** | 214                 | 133          | 166.51                     | 147.75       |
>
> The results show that both optimization leads to significantly less memory usage due to the sparsification of the dense matrix. As for training time, column selection alone yields a >4× speedup (214 -> 51 min) and >2× memory reduction (166.5 -> 71.9 MB) by computing only query-bit‑matched entries. Row selection only adds a smaller benefit (72 -> 51 min), as when POI Bloom filters have many bits, the PyTorch sparse‑dense matrix multiplication is not significantly faster than dense counterparts (due to memory access patterns). Thus, column selection provides the lion’s share of our efficiency gains, with row selection as a complementary, secondary optimization.
>
> **W3. Scalability and generalization.**
>
> We clarify that we have tested UrbanSparse's scalability on the GeoGLUE [1] benchmark (with 2.85 M POIs) in Appendix F.1. As shown in Table 10, UrbanSparse matches or slightly exceeds the strong multimodal baseline MGeo and trains in **54 min** versus **131 min** for DPR‑D :
>
> | Method          | Recall@10  | NDCG@5     | NDCG@1     | Training Time (min) |
> | --------------- | ---------- | ---------- | ---------- | ------------------- |
> | MGeo            | N/A        | N/A        | 0.5270     | —                   |
> | DPR‑D           | 0.7611     | 0.6318     | 0.5350     | 131                 |
> | **UrbanSparse** | **0.7621** | **0.6344** | **0.5310** | **54**              |
>
> We apologize for the lack of clarity in the main text and will add a concise reference in Section 5.5 to guide readers directly to these results.
>
> **Q3. Remaining inference bottlenecks**
>
> Thank you for pointing this out. Our detailed profiling shows that Bloom‑filter hashing adds negligible overhead. The primary cost arises from **our custom CUDA kernels for sparse representation calculations**: the non‑coalesced memory accesses in our kernel and an insufficiently optimized kernel dispatch strategy incur significantly higher latency than vendor‑optimized dense kernels from NVIDIA experts. Consequently, while we achieve a 50× reduction in parameters, the 4× QPS gain is bounded by these CUDA kernel inefficiencies. We anticipate that expert‑tuned CUDA kernels (particularly, leveraging shared memory with better parallelism) will further narrow this gap.
>
> [1] Li et al. GeoGLUE: A geographic language understanding evaluation benchmark. arXiv:2305.06545, 2023.

---

### Official Review · Reviewer_Xk76 · 2025-07-05

**Clarity:** 2
**Significance:** 2
**Originality:** 2
**Rating:** 3
**Confidence:** 1

**Summary:**

The paper introduces a unified approach to address both geo-spatial prediction and retrieval tasks in the context of Urban Computing. To enhance computational efficiency, the authors employ Bloom filter-based sparse encoding. They suggest breaking down textual data using a combination of n-gram and dictionary-based tokenizers, which are then transformed into Bloom filter representations via random hashing.

Additionally, a dense semantic codebook is utilized to capture fine-grained urban features. The method further proposes a joint optimization framework that merges the traditionally separate processes of geographical prediction and retrieval. Experimental results demonstrate that this the suggested model outperforms several baseline methods.

**Questions:**

Please see above

**Ethical Concerns:**

["NO or VERY MINOR ethics concerns only"]

**Final Justification:**

While the authors mention potential broader applications of their method, these extensions are well beyond the scope of the current paper, and there is no clear evidence provided to demonstrate their applicability or potential benefits. Therefore, in my view, the scope of the paper remains focused on a single urban computing application. That said, based on the responses of other reviewers who appear to be more domain-expert, it seems they find merit in the contribution.

Regarding to answer to W4 (Baseline comparisons) the authors claim in the rebuttal "While we fail to test ERNIE-GeoL (KDD 2022) and MGeo (SIGIR 2024) on our datasets due to reproducibility issue," – It's not clear to me what this means. It's also not clear why they place a "strong baseline" in the Appendix and not in the main paper.

Overall, while the rebuttal addresses some of my concerns through clarification, as a non-expert in this area, I remain uncertain whether the paper passes the acceptance threshold. So, I remain with my score, but I won't be disappointed if the paper is accepted.

**Limitations:**

Yes

**Paper Formatting Concerns:**

No formatting issues

**Quality:**

2

**Strengths And Weaknesses:**

**Strength:**
The paper tries to handle the task of unifying two domains of geo-spatial prediction and retrieval, in Urban Computing. Some ideas in the method, such as using of Bloom Filters, and unified optimization seem to be novel in this domain.

**Concerns and Weaknesses:**
One primary concern with this paper is IMO its niche application domain. It is not immediately evident how relevant or appealing the work will be to the broader NeurIPS community. The approach is highly tailored to Urban Computing, and the paper does not clearly articulate how the proposed method could generalize to other, more common NeurIPS-related domains.

Beyond that, several specific issues are noted:

1.	The proposed method comprises multiple interconnected components, which introduces significant complexity, IMO. This abundance of detail obscures the central contribution, making the core idea hard follow, in the Method section.
2.	A more thorough definition and explanation of Bloom filters would be helpful. Given that Bloom filters are prone to false positives (but not false negatives), it is important to clarify how this characteristic affects the system's performance and reliability

In terms of Evaluations:

3.	I believe, that the choice of datasets (Beijing and Shanghai) is not well justified. The paper should explain what datasets are commonly used in Urban Computing and why these two were selected.
4.	Regarding baseline comparisons, the retrieval methods used, e.g OpenAI2, BM25, BERT, etc., seems to be  all general-purpose models, not tailored or trained for Urban Computing. I am afraid that the lack of comparison with domain-specific baselines weakens the relevance of the evaluation.
5.	The authors justify excluding methods relying on GPS trajectory by stating that such data are unavailable for their datasets. However, this raises the question: why not use datasets employed by those other studies, especially if aiming for a fair and competitive comparison with recent work in the field?
5.	The paper lacks details on training, validation, and test splits. This is essential for reproducibility and to understand the robustness of the model.
6.	Although the paper proposes a unified framework, it does not provide evidence or ablation studies demonstrating its advantages over treating the prediction and retrieval tasks separately.

---

> ### Author Rebuttal · Authors · 2025-07-30
>
> Thank you for the careful reading and constructive feedback. We would like to address each weak point as follows:
>
> **Overall concern:** We clarify that Urban Computing is not a niche application; it has become a cornerstone of real‑world AI applications, powering services such as Google Maps, Uber, and food‑delivery platforms that billions rely on daily. The technical challenges studied in our work, i.e., fusing large‑scale text search with spatio‑temporal forecasting, also appear in areas such as temporal recommender systems and video‑event search. More importantly, our unique integration of Bloom‑filter‑based representation and two-level contrastive learning are not confined by urban computing. Such strategy can be applied to the general text-based retrieval/recommender systems, unlocking the joint benefits of retrieval and prediction and bringing broader impact to the NIPS community.
>
> **W1. Method clarity.**
> Thank you for your feedback. We will restructure Section 3 to center on our core innovation: the use of Bloom filters and tailored contrastive learning at both bit‑ and region‑level. We will also introduce a more intuitive figure to illustrate the overall framework, highlighting its core components and avoiding overwhelming training/optimization details.
>
> **W2. Bloom filters and false positives.**
> We apologize for the oversight in the main text. A comprehensive study is already provided in Appendix E.1, including a definition of Bloom filters and ablation experiments varying bit‑vector length (*m*) and number of hash functions (*k*), which show that false positives are empirically mitigated when $m \geq 8192$, $k \geq 2$. We will add a concise reference in Section 4.1 to guide readers directly to these results.
>
> **W3. Dataset choice justification.**
> Our primary goal is to explore the joint benefit of retrieval and prediction tasks, which requires datasets that support both. For this reason, we selected the Beijing and Shanghai benchmarks (from Liu et al. [2]), which provide over 160,000 realistic user queries paired with 120,000 POIs that uniquely enable industry-scale joint evaluation. In contrast, commonly used POI datasets (e.g., from OpenStreetMap or Foursquare) typically contain fewer than 30,000 POIs and lack user queries, making them unsuitable for retrieval tasks.
>
> However, we understand that evaluating on additional benchmarks can further demonstrate generality. Therefore, we also assess UrbanSparse on the widely used GeoGLUE \[1] benchmark, where UrbanSparse achieves retrieval performance comparable to strong PLM‑based models (see Appendix F.1). For population prediction, UrbanSparse also achieves the best $R^2 = 0.5344$, outperforming the next best method, CityFM ($R^2 = 0.4774$). Full results will be included in the revised manuscript.
>
> **W4. Baseline comparisons.**
> In addition to general‑purpose models, we have already compared UrbanSparse against domain‑specific methods. Specifically, we compare with the latest versions of POI-based prediction methods HGI (ISPRS 2023) and CityFM (CIKM 2024), and retrieval methods DrW (SIGMOD 2023). While we fail to test ERNIE-GeoL (KDD 2022) and MGeo (SIGIR 2024) on our datasets due to reproducibility issue, we found that both methods are adapted from DPR (EMNLP 2020) with geospatial components. Hence, we combine DPR with geospatial distance to form DPR-D, which we believe to be a strong baseline (As demonstrated in Appendix F.1, DPR-D performs better than MGeo on GeoGLUE).
>
> **W5. GPS-based methods and datasets.**
> Mobility-based methods often rely on multi-modal fusion, which is orthogonal to our primary objective of bridging retrieval and prediction. Datasets used in those works (e.g., NYC) contain rich mobility traces (>2,000,000 taxi records) but only a limited number of POIs (<30,000). Applying our method to such datasets would require major architectural changes to handle the mobility signal, which is beyond our current scope. For a fair comparison, we focus only on methods that use the same input modalities, leaving exploration of mobility-enhanced models to future work.
>
> **W6. Train/validation/test splits.**
> We apologize for the lack of detail in the main text. The full settings are described in Appendix D:
>
> * For retrieval, we adopt a fixed train/dev/test split of 0.81:0.09:0.10, following Liu et al. \[2], and make our data and splits publicly available.
> * For prediction, we follow standard unsupervised representation learning practice by evaluating embeddings with a scikit-learn RandomForestRegressor using 5-fold cross-validation across all urban regions.
>
>  We will bring these details into Section 4.3 for clarity and reproducibility.
>
> **W7. Ablation on unified vs. separate tasks.**
> Tables 2–4 already report “w/o individual” (i.e., training only on prediction) and “w/o collective” (i.e., training only on retrieval), showing clear performance degradation when either task is omitted. We will highlight these rows and the corresponding discussion to emphasize the necessity and effectiveness of joint training.
>
> We hope the revisions and additions can address your concern and improve the quality, robustness, and clarity of our work.
>
> \[1] Li et al. GeoGLUE: A geographic language understanding evaluation benchmark. arXiv:2305.06545, 2023.
>
> \[2] Liu et al. Effectiveness perspectives and a deep relevance model for spatial keyword queries. Proc. ACM on Management of Data, 1(1):1–25, 2023.

---

> > ### Comment · Reviewer_Xk76 · 2025-08-06
> > **post rebuttal**
> >
> > I would like to thank the authors for their effort to answer my concerns. Although the rebuttal answers some of my concerns with clarifications, as a non-expert in this domain I am still not sure if the paper passes the acceptance bar.

---

> > > ### Author Response · Authors · 2025-08-06
> > > **Follow-up: Request for Further Feedback and Suggestions**
> > >
> > > Thank you very much for your thoughtful reply! We understand that this topic may fall outside your core area, and we’re grateful that you still took the effort to review and discuss it with us.
> > >
> > > If there are any remaining concerns or areas where the paper still appears unclear or uncertain, we would be grateful if you could point them out. Your suggestions would be very valuable for helping us improve the work further, both during the revision process and beyond.

---

### Official Review · Reviewer_Yh1G · 2025-07-16

**Clarity:** 3
**Significance:** 2
**Originality:** 3
**Rating:** 4
**Confidence:** 4

**Summary:**

This paper introduces UrbanSparse, a novel method aimed at enhancing the accuracy of population density prediction, house price estimation, and point-of-interest (POI) retrieval by leveraging unified geospatial text encoding and learned embeddings. The approach uses Bloom-filter encoding to transform geographic text data into sparse representations, enabling efficient learning across different urban environments. By sharing useful information between tasks, UrbanSparse improves model performance without relying on complex sequential information.

The authors validate their method through extensive experiments conducted on datasets from Beijing and Shanghai. These experiments demonstrate that UrbanSparse outperforms existing methods in population density prediction and house price estimation while also showing promising results in POI retrieval tasks. Detailed descriptions of the experimental setup, including training algorithms and hyperparameter tuning processes, are provided to ensure reproducibility and facilitate further research.

In addition to presenting strong empirical results, the paper discusses the limitations of the current approach and suggests potential areas for future improvement. For instance, the authors consider how to handle noisy data or sparse inputs more effectively. They also explore possible applications and implications of their work, highlighting its potential impact on urban planning, resource allocation, and user-centric services. Overall, UrbanSparse not only advances computational efficiency and reduces resource consumption but also opens new avenues for addressing privacy concerns and fairness challenges in geospatial data analysis. Through thorough data analysis and theoretical exploration, the paper convincingly demonstrates the practical value and broad applicability of UrbanSparse in real-world scenarios.

**Questions:**

1、How does UrbanSparse handle extremely noisy or highly sparse inputs, and what strategies could be employed to improve performance under these conditions?

2、Can you provide a more thorough comparison with existing state-of-the-art techniques, highlighting both the advantages and limitations of UrbanSparse relative to other methods?

3、Could you simplify or provide additional background explanations for some of the advanced concepts in the paper to make it more accessible to a broader audience?

4、Have you considered the potential negative societal impacts of deploying UrbanSparse in real-world applications, such as bias in predictions or displacement effects in housing markets? What mitigation strategies do you propose?

**Ethical Concerns:**

["NO or VERY MINOR ethics concerns only"]

**Final Justification:**

The authors' responses have sufficiently addressed my concerns.

**Limitations:**

While the authors acknowledge some technical limitations of UrbanSparse—such as its sensitivity to sparse inputs and the need for further optimization—there is no explicit discussion of potential negative societal impacts associated with the deployment of their method in real-world settings. This is an important omission, especially given the application domains such as housing price estimation and population density prediction, which can have significant implications for equity, privacy, and public policy.

**Paper Formatting Concerns:**

There are no major format issues

**Quality:**

3

**Strengths And Weaknesses:**

Strengths：
The paper demonstrates high quality through its rigorous experimental setup and detailed documentation, ensuring reproducibility and credibility of the results. The methodology is articulated, making it easy for readers to understand how UrbanSparse integrates Bloom-filter encoding with learned embeddings for geospatial text processing. By showcasing practical applications in population density prediction, house price estimation, and POI retrieval on real-world datasets from Beijing and Shanghai, the authors highlight the significant potential of their approach in urban planning and resource allocation. This combination not only enhances computational efficiency but also opens new avenues for addressing complex geospatial data analysis challenges.

Weaknesses：
Despite its strengths, the paper could benefit from more robust validation under varied conditions, particularly in handling extremely noisy or highly sparse inputs, which limits its applicability in certain real-world scenarios. Additionally, while the technical details are well-documented, some advanced concepts might be challenging for non-specialist readers, suggesting a need for clearer explanations or simplified background information. Furthermore, a deeper comparison with existing state-of-the-art techniques would help clarify the unique advantages of UrbanSparse and strengthen its claim of originality. Addressing these areas would enhance the overall impact and broad applicability of the paper.

---

> ### Author Rebuttal · Authors · 2025-07-30
>
> Thank you for your constructive feedback. We would like to address each of the specific questions as follows:
>
> **Q1:** We clarify that handling extremely noisy or sparse inputs is a very different challenge out of the scope of this work. However, we agree that further validation would clarify UrbanSparse's robustness in such scenarios.
>
> To address this, we have conducted additional evaluations using the established GeoGLUE [1] benchmark, containing 2,849,754 POIs, with over 50% deliberately introduced fake POIs and queries with shuffled coordinates. Results demonstrate that UrbanSparse achieves competitive POI retrieval performance comparable to strong PLM-based methods (detailed results in Appendix F.1). For the population density prediction task under noisy conditions, UrbanSparse still delivers superior performance (\$R^2=0.5344\$) compared to the second-best method, CityFM (\$R^2=0.4774\$). We attribute this to the integration of Bloom filters and contrastive learning, which effectively mitigates the impact of noise through mining useful text bits and ignoring useless information. As for sparse input, we clarify that the Beijing dataset does contain regions with highly sparse POIs, and our method performs the best among all tested methods, indicating a certain degree of robustness. We plan to add the corresponding case studies in these data-sparse regions to clarify the detailed impact of sparse input.
>
> In addition, two straightforward strategies may be useful to strengthen robustness:
> (1) Integrating off-the-shelf query rewriting modules to reduce query noise.
> (2) Explicitly marking empty areas with special markers (e.g., random points) to better inform the model.
>
> We will present the full results and detailed ablation studies on these strategies in the revised Appendix.
>
> **Q2:** We have rigorously compared UrbanSparse with publicly available state-of-the-art baselines in both geospatial prediction and retrieval domains, including prominent methods such as HGI (IJGIS 2023), CityFM (CIKM 2024), and DrW (VLDB 2023). We omitted certain recent methods due to either reproducibility limitations or their exclusive reliance on specific data types (e.g., human-mobility data, taxi origin-destination pairs), which are not generally accessible for a fair comparison. In the revision, we will clearly articulate these points to better contextualize our comparisons.
>
> **Q3:** Thanks for your feedback. We will clarify advanced concepts such as Bloom filters, contrastive learning, and mutual information maximization. In the revised manuscript, these concepts will be thoroughly defined with intuitive explanations and complemented by visual illustrations to facilitate understanding among readers unfamiliar with these technical details.
>
> **Q4:** We recognize the importance of explicitly discussing potential negative societal impacts, especially considering the sensitivity of application domains like housing and urban planning. We will move the limitation section from the appendix to the main content, addressing possible issues such as prediction biases and unintended reinforcement of socioeconomic disparities.
>
> We hope our clarifications and additional experiments can address your concern.
>
> [1] Li et, al. Geoglue: A geographic language understanding evaluation benchmark. abs/2305.06545, Arxiv 2023.

---

> > ### Comment · Area_Chair_8CSf · 2025-08-08
> >
> > Dear Reviewer Yh1G,
> >
> > The author-reviewer discussion phase will end today. Please read the authors' rebuttal and check if they have addressed your concerns.
> >
> > Thanks,
> > AC

---

### Author Response · Authors · 2025-08-07
**Looking forward to further discussions**

Dear AC and Reviewers,

Thank you for your time and thoughtful feedback during the review process.

We would like to kindly follow up to ask whether our responses and additional experiments have adequately addressed your concerns. With only two days left in the discussion period, we remain open and eager to engage further. If you have any remaining questions or points that require clarification, we would be happy to provide further details or make revisions.

Thank you again for your constructive input and continued engagement.

Best regards,

Author of Submission #20849

---

### Decision · Program_Chairs · 2025-09-17

**Decision:**

Accept (poster)

**Comment:**

This paper proposes UrbanSparse, a new method that unifies geospatial prediction and retrieval through a sparse-dense representation architecture. By synergistically combining these tasks, UrbanSparse eliminates redundant systems while amplifying their mutual strengths.

Pros:
- The idea of unifying geospatial prediction and retrieval is interesting. The proposed method is novel
- The paper demonstrates high quality through its rigorous experimental setup and detailed documentation
- The paper is clearly written and easy to follow

Cons:

Most of the reviewers’ concerns are addressed, except for the one below.

- As pointed out by Reviewer Xk76, one primary concern with this paper is its niche application domain.  It is not immediately evident how relevant or appealing the work will be to the broader NeurIPS community. The approach is highly tailored to Urban Computing, and the paper does not clearly articulate how the proposed method could generalize to other, more common NeurIPS-related domains.

The paper can be accepted if Senor AC or Program Chairs think the topic fits NeurIPS well.